# The Risk of Cardiovascular Disease among Male and Female Participants Treated for Gout in the Multi-Ethnic Study of Atherosclerosis (MESA)

Surbhi Gupta †, Elizabeth Miller †, Sharon Stein Merkin, Maureen McMahon, Karol E. Watson and John D. FitzGerald *

Department of Medicine, UCLA David Geffen School of Medicine, Los Angeles, CA 90095, USA
* Correspondence: jfitzgerald@mednet.ucla.edu
† These authors contributed equally to this work.

**Abstract:** (1) The Multi-Ethnic Study of Atherosclerosis (MESA) is a multi-center longitudinal cohort study designed to investigate the risk factors associated with the incidence of CVD. The purpose of this study is to examine the impact of gout on incident CVD. (2) Participants reporting the use of gout-specific medications (urate lowering drugs or colchicine) were compared with non-users. Kaplan–Meier survival curves and multivariable models to control for known CV risk factors evaluated hazard ratios (HR) between participants taking gout medications versus those not taking gout medications. (3) For the 6734 participants, analyses were stratified by gender owing to a gout-gender interaction. For the 164 male and 59 female participants taking gout medications, Kaplan–Meier (unadjusted) survival curves demonstrate that participants taking gout medications have higher rates of CVD than participants not taking gout medications, particularly for women. After controlling for known CV risk factors, the adjusted HR for female participants taking gout medications was 1.79 (0.99, 3.23), $p = 0.05$; the adjusted HR for male participants on gout medications was 1.20 (0.81, 1.77), $p = 0.36$; (4) Participants treated for gout in this study have many comorbid conditions with known CVD risk factors, making it difficult to confirm the independent effect of gout on CVD. There are a paucity of data on women with gout. These findings suggest that there is a clinically meaningful and potentially greater risk of CVD among women with gout as compared to men.

**Keywords:** gout; cardiovascular risk; female

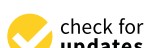



## 1. Introduction

Gout is the most common form of inflammatory arthritis, affecting 8.3 million adults in the United States [1]. While gout has afflicted people for centuries, its prevalence has doubled over the last 20 years due to an aging population, increased use of medications that contribute to hyperuricemia, and an increased prevalence of obesity and metabolic syndrome [2].

The association between gout and increased cardiovascular disease (CVD) was first reported in 1951 [3]. Since then, the gout-CVD association has been replicated in several large population studies. The MRFIT study, which controlled for traditional cardiovascular (CV) risk factors [4], and the Aspirin Myocardial Infarction Study, which controlled for renal function, diuretic usage, and the presence of metabolic syndrome [5], both associated the independent CV risk of gout with myocardial infarction (MI). Patients with gout also have higher rates of heart failure, as demonstrated by Krishnan and colleagues [6] and higher rates of CV death in the Health Professionals follow-up study [7]. Another study found in a cohort of patients that serum urate correlated with the degree of coronary artery calcification after adjusting for other risk factors, suggesting that hyperuricemia may also be an independent factor for coronary artery disease [8].

While gout is a disease that predominantly affects males, the prevalence of gout in females increases sharply after menopause [9]. Despite the roughly 2.2 million American women who suffer from gout, it is unclear if and how gout impacts women differently. Existing data suggest that gout-related musculoskeletal complaints, comorbid conditions, and functional impairment are more severe in women with gout compared to men [10]. Additionally, there is a growing body of data that suggests a higher risk for acute myocardial infarction (MI) attributable to gout in women than in men [11–13].

Despite existing data, unanswered questions remain. It is uncertain whether the gout-CVD association is independent of CV risk factors and whether this association may be modified by gender, age, or race.

The aims of our study are to evaluate the impact of gout on the incidence of CVD in women and men from various backgrounds using the Multi-Ethnic Study of Atherosclerosis (MESA) longitudinal cohort.

## 2. Methods

### 2.1. Study Population

MESA is a multi-center longitudinal cohort study designed to investigate the risk factors associated with the incidence of CVD and progression of subclinical CVD in four racial/ethnic groups (Caucasians, African Americans, Chinese Americans, and Hispanics). A detailed description of MESA has been published previously [14]. Participants included 6814 men and women aged 45–84 years recruited from six field centers (Baltimore, MD, USA; Chicago, IL, USA; Forsyth County, NC, USA; Manhattan, NY, USA; Los Angeles, CA, USA; and St. Paul, MN, USA) between the years 2000 and 2002 and free of clinical CVD at enrollment. Similar numbers of men and women were recruited based on pre-specified age and race/ethnicity quotas. The study followed participants via phone, mail, and in-person interviews for a median of 12 years. Five total examinations, including a baseline, were administered during the time of this analysis: exam 1 (July 2000–July 2002), exam 2 (September 2002–February 2004), exam 3 (March 2004–September 2005), exam 4 (September 2005–November 2007), and exam 5 (April 2010–December 2011).

All MESA participants provided informed consent. The MESA study protocol was reviewed and approved by each of the institutional review boards of the six originating field centers [14].

### 2.1.1. Identification of Participants Treated with Medications Used Primarily for Gout

The diagnosis of gout is not available in MESA. However, outpatient medication use is extracted from a patient-filled questionnaire at each visit. Therefore, as a proxy for a diagnosis of gout, we selected participants who reported taking medications prescribed primarily to treat gout (allopurinol, probenecid, and colchicine). No febuxostat users were identified in this MESA sample. Other authors have used medication use as a method to enhance the specificity of ICD claims-based diagnoses [15–19]. With our classification strategy, participants with an unobserved diagnosis of gout who have not been treated with any of the above medications become part of the larger comparator group.

Documentation of the above medications at any of the five visits qualified for inclusion. Almost half of participants in our gout medication group were on one of the above medications at visit 1, with similar numbers of first-time documentation of medication exposure across the remaining visits.

The majority (172) of the identified 223 participants (77%) reported allopurinol use, 3 participants reported probenecid use, and 45 participants reported colchicine use without documentation of urate-lowering medications (3 with chronic use of colchicine in at least 4 of the 5 reporting periods and 42 with intermittent use in 3 or fewer reporting periods). We tested for a differential CVD effect by class of drug (ULT vs. colchicine without ULT). As there was no differential effect, we do not report drug-specific CVD associations.

Finally, to evaluate if there was a differential association with CVD for persistent vs. intermittent use of gout medications, we defined persistent use as continued documentation of gout medications through the remaining study periods once first documented, and patterns of non-persistence as either intermittent use or cessation. While intermittent use was frequent in this cohort as expected based on previously observed low adherence rates among patients with gout [20], there was no differential effect by type of persistence, so this persistence was not included in the final models.

### 2.1.2. Comparator Group

The remaining 6511 MESA participants without any exposure to gout medications were included as the comparator group.

### 2.2. Clinical End Point Assessments

Our primary CV clinical endpoint used the MESA-defined all causes of CVD (myocardial infarction, resuscitated cardiac arrest, definite/probable angina, stroke, or deaths related to stroke, coronary heart disease, or other CVD). Hospital records were obtained for an estimated 95% of hospitalized CV events and 96% of outpatient diagnoses. To retain a plausible causal pathway, for participants treated with gout medications, we required exposure to these medications to predate the CV event.

### 2.3. Covariate Data

Information about age, gender, race/ethnicity, medical history, and medication history was obtained using questionnaires and a detailed physical exam, performed at baseline and at follow-up visits. Information about hospitalizations, CV events, and CV deaths was obtained from the participants (or their families in the event of a CV death) and verified using medical records.

### 2.4. Statistical Analysis

Participants identified as being treated for gout (defined above) were compared to the remainder of the cohort with regard to baseline demographic, racial/ethnic status, and clinical covariates. A priori, based on existing literature [11], we tested the following variables for potential interactions with the gout-CVD association: age, gender, and race, using three separate multivariable models that included interaction terms with those variables. Only the gender-gout interaction term met our screening inclusion criteria ($p = 0.10$) for stratified analysis. Therefore, the following models were stratified and reported results based on gender.

Student's *t*-tests and chi-square tests were used to test for statistically significant differences by gout treatment. Unadjusted Kaplan–Meier curves were created to illustrate CVD events by treatment status and gender (Figure 1).

Multivariable Cox proportional hazard models sequentially adjusted gout-CVD associations for age (years), race/ethnicity (Caucasian, Chinese American, African American, and Hispanic), foreign-born status, body mass index (kg/m$^2$), ASCVD risk estimate, and chronic kidney disease (CKD stage > 3). ASCVD is a CVD risk estimator derived from a formula that includes age, gender, race, blood pressure, cholesterol, history of diabetes, tobacco use, and whether a person is taking aspirin, statins, or hypertensive medication. For ease of interpretation of the ASCVD estimate, the variable was converted from a 0–1 to a 0–100 scale.

Missing data on main covariates led to the exclusion of 80 participants from this analysis, including those with documented use of gout medication only after an incident CVD event ($n = 4$), those missing follow-up information ($n = 7$), status on origin of nationality ($n = 22$), Atherosclerotic Cardiovascular Disease (ASCVD) data ($n = 35$), and information on hydrochlorothiazide (HCTZ) use ($n = 12$). This left a total of 6734 in the analytic sample (3176 men and 3558 women).

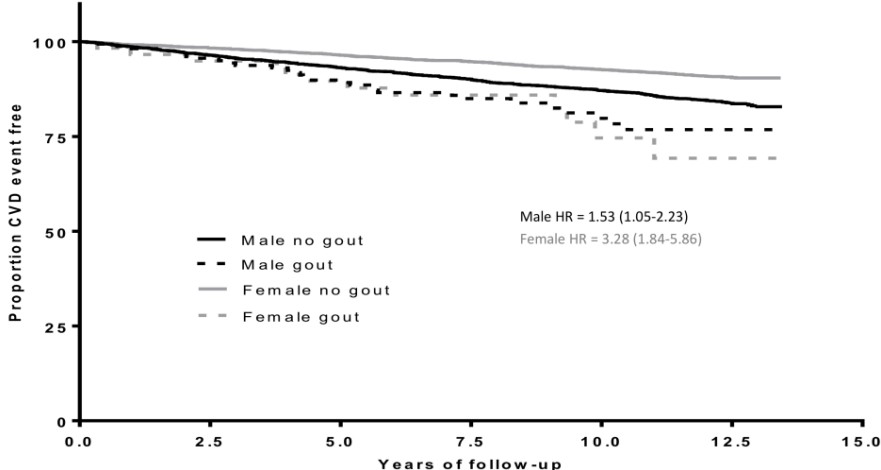

Legend:

| Years of Follow-up | 2.5 | 5.0 | 7.5 | 10.0 | 12.5 | 15.0 |
|---|---|---|---|---|---|---|
| Male, No gout | 3012 | 2810 | 2588 | 2428 | 2262 | 895 |
| Male, gout | 164 | 152 | 136 | 95 | 56 | 17 |
| Female, No gout | 3499 | 3347 | 3128 | 2995 | 2861 | 1296 |
| Female, gout | 59 | 55 | 49 | 40 | 18 | 4 |

**Figure 1.** Unadjusted cardiovascular disease (CVD) event-free survival curves for participants treated vs. not treated for gout by gender.

## 3. Results

### 3.1. Baseline Descriptors

Male and female participants treated for gout were significantly older, more obese, had poorer renal function (a higher proportion with CKD > 3), and had higher rates of HCTZ use than their counterparts. In addition, both male and female participants treated for gout had higher baseline ASCVD scores and higher baseline IL-6 levels (Table 1). Among female participants, those treated for gout were more likely to be postmenopausal than female participants not treated for gout. There were similar rates of HRT use between the two groups (Table 1).

### 3.2. CVD Outcomes

Among the 164 male participants treated for gout, there were 29 CVD events (2.15 events per 100 person-years, *p* = 0.02); among the 3012 male participants not treated for gout, there were 441 CVD events (1.41 events per 100 person-years, *p* = 0.02). Among the 59 female participants treated for gout, there were 12 CVD events (2.15 events per 100 person-years), *p* < 0.0001; among the 3499 female participants not treated for gout, there were 299 CVD events (0.78 events per 100 person-years) (Table 1). Figure 1 shows the CVD event-free probability for males and females separated by gout status. The unadjusted HR of gout treatment status on CVD among men was 1.53 (95% CI: 1.05–2.23), and the unadjusted HR of gout treatment status among women was 3.28 (95% CI: 1.84–5.86).

To examine the independent association of gout treatment status on CVD, multivariable analysis controlled for age, gender, race, country of origin, ASCVD risk estimator, BMI, and presence of CKD stage 3 or worse (Table 2). The HR of CVD among participants treated for gout was 1.30 (0.94, 1.80, *p* = 0.11), similar to that reported in other studies, though not significant after controlling for the above covariates. Among women, the adjusted HR was higher: 1.79 with a 95% CI that just crossed the null hypothesis (0.99, 3.23, *p* = 0.05). Among men, the adjusted HR was 1.2 (0.81, 1.77, *p* = 0.36).

**Table 1.** Unadjusted demographics of male and female participants by gout treatment status.

| | Male | | | Female | | |
|---|---|---|---|---|---|---|
| | Treated for Gout (*n* = 164) | Not Treated for Gout (*n* = 3012) | *p* Value | Treated for Gout (*n* = 59) | Not Treated for Gout (*n* = 3499) | *p* Value |
| Demographic variables | | | | | | |
|   Mean Age in years (s.d.) | 65 (9) | 62 (10) | 0.0001 | 68 (8) | 62 (10) | <0.0001 |
|   Self-identified Race (%) | | | 0.02 | | | 0.01 |
|     Caucasian-American | 37 | 39 | | 46 | 38 | |
|     Chinese-American | 16 | 12 | | 10 | 12 | |
|     African-American | 35 | 26 | | 37 | 28 | |
|     Hispanic-American | 12 | 23 | | 7 | 22 | |
|   Foreign-born status (%) | 26 | 32 | 0.02 | 19 | 32 | 0.01 |
| CV Risk Variables | | | | | | |
|   Mean ASCVD Risk Score (s.d.) | 0.21 (0.14) | 0.16 (0.13) | <0.0001 | 0.18 (0.14) | 0.11 (0.13) | <0.0001 |
|   Mean BMI (s.d.) | 30.0 (4.7) | 27.7 (4.4) | <0.0001 | 32.6 (7.3) | 28.7 (6.2) | <0.0001 |
| Risk Factors for Gout | | | | | | |
|   CKD > 3 n (%) | 56 (34%) | 330 (11%) | <0.0001 | 20 (34%) | 467 (13%) | <0.0001 |
|   HCTZ n (%) | 20 (12%) | 180 (6%) | 0.001 | 11 (19%) | 274 (8%) | 0.002 |
| Inflammatory Mediators | | | | | | |
|   Mean C-reactive protein | 3.5 | 2.9 | 0.2 | 5.5 | 4.5 | 0.21 |
|   Mean Interleukin-6 | 1.9 | 1.5 | <0.0001 | 2.0 | 1.6 | 0.02 |
| Post-menopausal (%) | N/A | N/A | N/A | 98 | 90 | 0.02 |
| Hormone replacement (%) | N/A | N/A | N/A | 30 | 32 | 0.11 |
| Total CVD Events n (%) | 29 (17.6%) | 441 (14.6%) | 0.05 | 12 (20.3%) | 299 (8.5%) | 0.003 |
| CVD events per 100 person-years | 2.15 | 1.41 | 0.02 | 2.46 | 0.78 | <0.0001 |

ASCVD score: 10-year risk of developing CVD based on the ASCVD risk parameter estimate (19). CKD: reported as the number of participants with CKD if GFR < 60. HCTZ use: reported number of participants on HCTZ at any time point throughout the study.

**Table 2.** Multivariable cardiovascular risk adjusted hazard ratios for participants.

| | All Participants (*n* = 6734) | | | Female (*n* = 3558) | | | Male (*n* = 3176) | | |
|---|---|---|---|---|---|---|---|---|---|
| Parameter | Hazard Ratio | 95% Confidence Limits | *p* Value | Hazard Ratio | 95% Confidence Limits | *p* Value | Hazard Ratio | 95% Confidence Limits | *p* Value |
| Gout | 1.30 | (0.94, 1.80) | 0.11 | 1.79 | (0.99, 3.23) | 0.05 | 1.20 | (0.81, 1.77) | 0.36 |
| Male gender | 1.61 | (1.39, 1.87) | <0.0001 | | | | | | |
| Age (years) | 1.030 | (1.02, 1.04) | <0.0001 | 1.04 | (1.02, 1.06) | <0.0001 | 1.02 | (1.01, 1.04) | 0.0008 |
| Race/Ethnicity | | | | | | | | | |
| Caucasian referent group | 1.0 | | | 1.0 | | | 1.0 | | |
| African-American | 0.92 | (0.77, 1.10) | 0.38 | 0.79 | (0.60, 1.05) | 0.10 | 1.01 | (0.81, 1.27) | 0.92 |
| Chinese | 0.94 | (0.67, 1.32) | 0.71 | 0.95 | (0.55, 1.63) | 0.84 | 0.92 | (0.59, 1.42) | 0.69 |
| Hispanic | 1.15 | (0.91, 1.44) | 0.23 | 0.94 | (0.64, 1.38) | 0.75 | 1.29 | (0.97, 1.70) | 0.079 |
| Birth origin status (Foreign vs. US born) | 0.71 | (0.56, 0.89) | 0.004 | 0.77 | (0.53, 1.13) | 0.18 | 0.67 | (0.50, 0.90) | 0.009 |
| Body Mass Index (kg/m$^2$) | 1.02 | (1.01, 1.04) | 0.002 | 1.04 | (1.02, 1.06) | 0.0001 | 1.01 | (0.98, 1.03) | 0.54 |
| ASCVD | 1.03 | (1.02, 1.03) | <0.0001 | 1.03 | (1.02, 1.04) | <0.0001 | 1.03 | (1.02, 1.04) | <0.0001 |
| Chronic Kidney Disease ≥ 3 | 1.14 | (0.95, 1.37) | 0.16 | 1.14 | (0.87, 1.51) | 0.34 | 1.12 | (0.87, 1.44) | 0.39 |

## 4. Discussion

Using the MESA cohort, the gout-CVD associations reported here support prior published reports on the gout-CVD association, but most notably for women, with an adjusted HR of 1.8 (*p* = 0.05) and a 95% confidence interval (95% CI) just overlapping the null hypothesis (0.99, 3.23). De Vera and colleagues reported a relative risk (RR) of 1.4 (95% CI, 1.2, 1.6) for all myocardial infarctions from a Canadian longitudinal, integrated healthcare database [11]. De Vera and colleagues further reported that the female RR of 1.4 was significantly higher (*p* < 0.005 of interaction) than the male RR of 1.1.

Due to MESA's recruitment efforts targeting an older population with equal proportions of gender, the sample is enriched for women treated for gout, representing 26.5% (59/223) of the participants treated for gout. This representation contrasts the recruitment

of women to clinical trials. In an analysis of 24 gout clinical trial publications between 2000 and 2021, women represented only 12.5% of recruited subjects [21].

The observed increased gout-CVD HR for women in this cohort suggests that gout or hyperuricemia may be an additional independent CV risk among women. The larger gout-CVD association in women is potentially influenced by several factors. Attributable to the favorable effect of pre-menopausal estrogen on the fractional excretion of serum urate, gout is less frequently experienced in women before menopause. However, when present in women, gout flares last longer, require higher use of anti-inflammatories, and result in more frequent ER visits and hospitalizations compared to men [10]. In addition to hyperuricemia and gout, menopause brings with it an increased risk of CVD [22], with women having more unstable plaques than men [23,24].

Additional studies have shown that women with asymptomatic hyperuricemia not only have an increased association with CVD but also with renal comorbidities and all-cause mortality [25,26]. Unfortunately, historical clinical trials of urate-lowering medications have had underrepresentation of females, with participation to prevalence ratios of 0.34 (for hyperuricemia) and 0.69 (for gout) that have only worsened over the last 15 years [21]. Therefore, there remains a need to include more women in future studies to better understand the intersection of the female gender, hyperuricemia, and CVD.

From our MESA data, while the male and overall HR (1.2 and 1.3, respectively) are similar to other published reports [27,28], these associations did not remain statistically significant after controlling for known CV risk factors, which may be attributable to the smaller number of participants with gout identified in MESA.

Many authors have reported that the gout-CVD association remains significant after adjusting for demographic and CV risk factors [4,7,29]. There has been much postulation about the causal association between gout and/or hyperuricemia and CVD, which may be due to the known oxidative stress of xanthine oxidase (XO) [30]. XO is the cardinal enzyme in the purine metabolism pathway that generates uric acid by oxidizing hypoxanthine to xanthine and xanthine to uric acid. Oxidative stress is generated from the process, which is a major cause of endothelial dysfunction. Expression of XO is upregulated by stress, inflammation, and ischemia; levels of XO have been found to be elevated in those with coronary artery disease and heart failure. Allopurinol, an inhibitor of XO commonly used as a urate-lowering therapy for gout, has been shown to reduce oxidative stress and improve endothelial dysfunction in patients with CVD, diabetes, and hypertension [30].

Uric acid crystals may also directly contribute to cardiovascular pathology. An abstract from Nishimiya et al. reported in cadaver coronary arteries with a history of CVD demonstrated that those with a history of gout had significantly more uric acid crystals in their coronary plaques, as well as a greater degree of intimal thickening [31].

Gout and cardiovascular disease may also share similar pathways of inflammation. This is suggested by the Canakinumab Anti-inflammatory Thrombosis Outcome Study (CANTOS) and Colchicine Cardiovascular Outcomes Trial (COLCOT). In CANTOS, patients with a history of prior MI and elevated C-reactive protein who received the interleukin-1 β inhibitor canakinumab had a 15% reduction in major cardiovascular events, with a concomitant 55% reduction in incident gout flares (among patients without gout) and a 31% reduction in gout flares (among patients without gout) [32,33]. In COLCOT, patients with recent MI who received low doses of daily colchicine had a 23% relative risk reduction in major vascular events [34].

The strength of this analysis is that MESA is a large prospective cohort study designed to study the incidence and progression of CVD that prospectively collected risk factors for CVD and validated major cardiovascular events through thorough follow-up of participants for over 12 years. CVD events are carefully adjudicated, as is a detailed collection of CV risk factors. Additionally, MESA has a significant number of older women, enhancing the number of women with gout available for study as compared to the previous literature.

However, there are several limitations to this analysis. Though MESA is a large cohort study, the proportion of participants identified as being treated for gout is small, and the

number of participants with CVD events who were treated for gout is even smaller. Our analysis is further limited in that neither the diagnosis of gout nor uric acid measurements were documented, necessitating the use of gout medication as a surrogate.

There are a number of important considerations to acknowledge with this approach. First, as participants with gout were identified solely by medication use, misclassification bias may exist as subjects with gout but not taking urate lowering therapy or colchicine were left in the comparator group, thereby attenuating any observable gout-CVD association. Conversely, not all patients who are prescribed urate-lowering therapies or colchicine have gout. An observational study from a single pharmacy found that of the 74 patients prescribed xanthine oxidase inhibitors, the indications for treatment were gout in 60.8%, asymptomatic hyperuricemia in 33.8%, and uric acid nephrolithiasis in 5.4% [35].

Twenty-one percent of our defined participants with gout medications had (primarily intermittent) colchicine use without ULT use. We looked for a differential association with CVD between the colchicine without ULT group and the ULT group (with or without colchicine use) and found none. Both medication groups had 18% CV events.

Colchicine has non-gout indications, including acute inflammatory calcium pyrophosphate disease (CPPD), recurrent or chronic pericarditis, and familial Mediterranean fever (FMF). While recurrent or chronic pericarditis and FMF are likely rare enough conditions to be unlikely to create misclassification, CPPD is relatively common. Furthermore, patients with FMF, chronic inflammatory CPPD, or other chronic auto-inflammatory conditions would likely be on chronic colchicine, which was infrequently observed (3 patients) in the MESA study (described above). There are fewer studies on CPPD and CVD, but a recent cohort study demonstrated a weak CPPD-CVD association (HR 1.32, 95% CI 1.01–1.73) [36].

As noted above, the number of participants expected to be treated for gout matches the observed cases in this cohort when compared to prevalence data from the literature. Based on an expected male prevalence of gout of 5.9% (all ages) to 12.2% (65+ years of age) and an expected female prevalence of 2.2% (all ages) to 3.0% (65+ years of age) [1,9], and a 50% prescription use of urate lowering therapy for patients with gout, we would expect 94–193 male gout participants in the MESA cohort (164 observed) and 40–54 female gout participants (59 observed).

We anticipate a similar number of participants with untreated gout, based on the expectation that 50% of gout patients are untreated with urate-lowering therapy [37–39]. Therefore, the comparator group in our analysis likely includes participants with untreated gout. This misclassification would bias the results towards the null. However, the proportion of these untreated gout participants (and therefore the potential bias) would be small in comparison to the size of the total comparator group (roughly 4% based on the assumptions above). Additionally, HR estimates have known limitations as they are approximations but not equivalent to relative risks. If risk changes over time, a summary HR estimate will not capture those variations. (The Kaplan–Meier analysis in Figure 1 does not suggest significant temporal variation in risk.)

Furthermore, other authors have reported that untreated gout patients have higher rates of CVD than patients treated for their gout [29]. On the other hand, as mentioned previously, COLCOT found that the use of low-dose colchicine in patients with recent MI had a 23% relative risk reduction in major vascular events [34]. Both of these limitations bias the HR estimate towards the null hypothesis. As a corollary, identifying patients treated with allopurinol or colchicine could bias the estimate towards the null by selecting patients on medications that may be associated with reduced CV risk based on the Cardiovascular Safety of Febuxostat or Allopurinol in Patients with Gout (CARES) [40] or COLCOT [34] studies, respectively.

Beyond potential misclassification, different choices to manage gaps in gout medication use (adherence) may have impacted modeling outcomes. We chose to define any gout medication exposure as probable gout without making assumptions about disease duration or incident gout. As an alternative, rather than excluding the 4 patients with CV events prior to gout medication documentation, these patients could have been included

in the comparator group (without gout medication exposure) up until their CV-defining event. However, we felt this implied a higher level of certainty about incident gout than the data provided. For that same reason, we did not left-censor observations where gout medication exposure was documented only in later study windows. Rather, we assigned participants with any gout medication exposure to the exposure group and those without any gout medication exposure to the comparator group.

Despite MESA being a large population cohort of older Americans, the number of participants treated for gout and participants treated for gout with CV events remained small, particularly for women, and therefore this study remains marginally powered.

For our multivariate Cox proportional analysis, based on the exposure definitions above and the decision on managing gaps in gout medication use, both exposure and comparator participant observational periods began with study entry and continued until the CV-defining endpoint or end of the observational study window.

If the CV risk changed differentially with time on study, this could bias the estimated hazard ratio between groups. This could have biased the result towards the null if CV risk was increasing for the longer duration of the study and away from the null if the risk was decreasing over time. Additionally, there is potential for immortal time bias due to the fact that study participants in the gout group needed to remain on study long enough to obtain a gout diagnosis.

Finally, participants reporting gout medication use were included in the study from 2000 to 2011. During a similar period (from 1990 to 2013), other authors report that annual age-adjusted cardiovascular mortality rates have declined by 22% from 376 to 274 per 100,000 [41]. This fall has been attributed to changes in primary and secondary prevention.

## 5. Conclusions

Gout is a relapsing inflammatory state resulting from the chronic deposition of monosodium urate crystals. The association of gout with MI and/or CVD events has been well documented in men, but there has been limited data for women with gout. Next, our findings support the observation that gout patients have more CVD events than patients without gout, largely supported by the higher prevalence of CV risk factors among male participants treated for gout. The stronger gout-CVD association among women is partly explained by known CV risk factors. The small number of participants identified as having gout limits the strength of this finding and warrants further investigation.

**Author Contributions:** Conceptualization, E.M. and J.D.F.; methodology, S.S.M. and J.D.F.; software, S.S.M.; validation, S.S.M., K.E.W. and J.D.F.; formal analysis, S.S.M. and J.D.F.; investigation, E.M., S.S.M., M.M., K.E.W. and J.D.F.; resources, K.E.W. and J.D.F.; data curation, S.S.M.; writing—original draft preparation, E.M. and J.D.F.; writing—review and editing, S.G., M.M., K.E.W. and J.D.F.; visualization, J.D.F.; supervision, J.D.F.; project administration, J.D.F.; funding acquisition, K.E.W. and J.D.F. All authors have read and agreed to the published version of the manuscript.

**Funding:** This research was supported by Claude D. Pepper Older American Independence Centers, contracts HHSN268201500003I, N01-HC-95159, N01-HC-95160, N01-HC-95161, N01-HC-95162, N01-HC-95163, N01-HC-95164, N01-HC-95165, N01-HC-95166, N01-HC-95167, N01-HC-95168 and N01-HC-95169 from the National Heart, Lung, and Blood Institute (NHLBI), and by grants UL1-TR-000040, UL1-TR-001079, and UL1-TR-001420 from NCATS. The authors thank the other investigators, the staff, and the participants of the MESA study for their valuable contributions. A full list of participating MESA investigators and institutions can be found at http://www.mesa-nhlbi.org (accessed on 18 November 2022).

**Institutional Review Board Statement:** Not applicable.

**Informed Consent Statement:** Not applicable.

**Data Availability Statement:** 20]MESA is funded by NHLBI (noted above). MESA Study data can be requested through the BioLINCC website at https://biolincc.nhlbi.nih.gov/ (accessed on 18 November 2022) following instructions from the BioLINCC_Handbook_V3.7, which can be found at https://biolincc.nhlbi.nih.gov/media/guidelines/handbook.pdf?link_time=2023-05-08_11:15:45.2 82360#page=15 (accessed on 18 November 2022).

**Acknowledgments:** The authors would like to thank Arun Karlamangla for his input on methodological design and David Elashoff for his input on our statistical analysis.

**Conflicts of Interest:** The authors declare no conflict of interest. The funders had no role in the design of the study; in the collection, analyses, or interpretation of data; in the writing of the manuscript; or in the decision to publish the results.

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
