# Peer review of "The Risk of Cardiovascular Disease among Male and Female Participants Treated for Gout in the Multi-Ethnic Study of Atherosclerosis (MESA)"

_2813-4583, doi:10.3390/gucdd1020009_

Round 1

Reviewer 1 Report

In this manuscript, the authors examine the association between gout diagnosis and cardiovascular risk.  To do this they use the MESA—multiethnic study of atherosclerosis data set.  They conclude that gout patients of both sexes are more likely to develop a cardiovascular diagnosis than those without gout, although this association disappears overall after correction for other cardiovascular risk factors.  However, among women, the relationship is both larger, and more likely to persist after adjustment (borderline significance).  This study supports a general trend in the literature that links gout with CV risk and therefore adds to the growing recognition of gout as a possible CV risk factor.  The study has a number of strengths, including the use of the MESA study data, which by design allows for accurate assessment of CV outcomes.  The fact that the original study enrolled roughly equal numbers of women (though fewer female than male gout patients, as is to be expected) is also quite valuable, and continues to add to the literature is also quite significant, and adds to the growing literature that at least directly, gout may have greater adverse impacts on women than men.  The study also has a number of limitations, mostly as noted by the authors, including the relatively small number of gout patients included, which despite the overall large cohort may have limited the ability to achieve significance in results where trends seem to be apparent.

The main limitation of the study, as noted by the authors themselves, is the definition of gout used as the entry criterion for the gout arm.  As the authors note, they did not have gout diagnoses available and so used gout treatment as a surrogate for gout diagnosis.  While this is probably the best they could do, I think this limitation may deserve additional comment in the discussion, at least.

Specific points:  

1)    The use of gout medications for defining gout has limitations, as noted above.  Among these are the fact that many patients with gout may not be treated, as noted by the authors.  In fact, based on ACP guidelines, most patients with gout do not received chronic urate lowering or anti-inflammatory therapy.  The authors statement that 50% of gout patients take urate-lowering therapy is likely a gross overstatement—that number more likely relates to the poor compliance rate when allopurinol is prescribed (which is actually even lower in many studies). It would be helpful if the authors would provide a broader discussion of the literature  as to the percent of gout patients actually receiving therapy.  As the authors themselves note, inclusion of untreated gout patients biases against the outcome so this concern does not impugn the integrity of the study.

2)    Again, the use of gout medications for defining gout has limitations, to the extent that some of these medications are used for other purposes, and/or inappropriately. The authors should comment on the possibility that some patients received urate lowering therapy not for gout but for asymptomatic hyperuricemia—a related but different condition.  Similarly, some patients receiving colchicine may have been taking it for pseudogout or pericarditis.  The authors should comment in their discussion at least, as to the level of certainty that patients taking gout medications actually have gout. 

3)    The possibility that the data presented underestimate the impact of gout on CV disease, specifically because of the possibility that gout treatments may have reduced CV outcomes (together with the fact that untreated gout patients are in the control group), bears additional discussion.  In particular, as the authors note multiple large controlled trials in the general CV risk population, and several retrospective studies in the gout population suggest that colchicine may reduce CV risk by roughly 25% or more. The possibility that the authors’ data   underestimate gout cardiac risk because the patients are treated should not go unremarked on.  I believe it is possible that a bigger effect would have been seen if the authors could have followed an untreated gout population, including significance in both the male and female populations.

4)    The authors’ efforts to correct for other interrelated risk factors are laudable, but it deserves mention that these efforts may have represented in an overcorrection.  For example, while kidney disease is certainly an in independent risk factor for CV disease, multiple studies have suggested the possibility that gout may be a CKD risk factor—so the variables are entangled in complex ways.  It deserves at least brief mention that the correction for some variables may actually remove the effect of mediating pathways through which gout might act to promote cardiovascular disease--again, biasing the study against its actual outcome and perhaps leading to an underrecognition of the impact of gout.

Additional questions:

1)    It would be helpful to specify the frequency/schedule of follow up visits in the study.

2)    The authors comment that they saw no differences in outcomes between patients who were compliant and/or on consistent treatment compared with those who were non-compliant and/or on intermittent treatment, and therefore they do not provide outcomes according to these groups.  However, these are potentially important observations; it would be helpful to have at least some data provided.

Author Response

We thank the reviewers for their thoughtful comments.

We have organized our responses by reviewer order.

Where similar comments, we responded in order of reviewer and subsequently reference.

Reviewer comments are pasted verbatim, flush with left margin.

Our responses are indented with references to updates in the manuscript either by location or pasted “quotations”.

In summary, we have primarily made edits (some extensive) to the methods and discussion sections as suggested by the reviewers. The results section is unchanged.

Reviewer 1

The main limitation of the study, as noted by the authors themselves, is the definition of gout used as the entry criterion for the gout arm.  As the authors note, they did not have gout diagnoses available and so used gout treatment as a surrogate for gout diagnosis.  While this is probably the best they could do, I think this limitation may deserve additional comment in the discussion, at least.

This comment was raised by all 3 reviewers and deserves greater specificity in the manuscript. As described in the manuscript, given the absence of gout diagnosis, as proxy, we identified patients using medications primarily used to treat gout.

To address reviewers’ comments, we have reworded our methods section to be more precise starting with the header. We have expanded our discussion section. (See below for detail.)

“2.2 Identification of participants treated with medications used primarily for gout

Diagnosis of gout is not available in MESA. However, outpatient medication use is extracted from a patient-filled questionnaire at each visit. Therefore, as proxy for a diagnosis of gout, we selected participants who reported taking medications prescribed primarily to treat gout (allopurinol, probenecid, and colchicine). No febuxostat users were identified in this MESA sample. Other authors have used medication use as a method to enhance the specificity of ICD claims-based diagnoses [15-19]. With our classification strategy, participants with an (unobserved) diagnosis of gout and not treated with any of the above medications, become part of the larger comparator group.”

We have opted to retain our original descriptor “participants treated for gout” rather than other options (e.g. “participants treated for likely gout” or “participants treated with medications used for gout”) for simplicity as the repeated use of longer descriptors could complicate the primary message.

Specific points:  

1)    The use of gout medications for defining gout has limitations, as noted above. Among these are the fact that many patients with gout may not be treated, as noted by the authors.  In fact, based on ACP guidelines, most patients with gout do not received chronic urate lowering or anti-inflammatory therapy.  The authors statement that 50% of gout patients take urate-lowering therapy is likely a gross overstatement—that number more likely relates to the poor compliance rate when allopurinol is prescribed (which is actually even lower in many studies). It would be helpful if the authors would provide a broader discussion of the literature  as to the percent of gout patients actually receiving therapy.  

The exact proportion of gout patients or gout patients with indications for ULT varies by study (each with limitations in design). Based on our review of the literature, we feel the 50% estimate is reasonable. We provide the additional requested detail here for the reviewers. We considered this text for the discussion, but ultimately opted to exclude as the detail on ULT use among gout patients, while relevant, felt tangential to the focus of the article. We defer to editor / reviewers if inclusion benefits the manuscript.

In a meta-analysis of 8 studies focusing on the care of patients with gout in primary care settings (across several countries) found that less than 50% of patients with gout were prescribed ULT. Using National Health and Nutrition Examination Survey (NHANES) from 2007-2010 Juraschek and colleagues estimated 7.7 million US adult patients with gout, 4.5 million with an indication for ULT (defined as currently on ULT or CKD 2-5 or nephrolithiasis – regardless of ULT use). Of these 4.5 million with an indication for ULT, 2.2 million (49.6%) were on ULT. Separte authors using (NHANES) from 2007-2014 reported 33% of patients with gout receive ULT, which was stable over the years. (Similar to Jurascheck analysis when the denominator is all patients with gout.)

As the authors themselves note, inclusion of untreated gout patients biases against the outcome so this concern does not impugn the integrity of the study.

We agree with the reviewer that the bias is towards the null. This is stated in the manuscript in the subsequent paragraph.

“Therefore, the comparator group in our analysis likely includes participants with un-treated gout. This misclassification would bias the results towards the null.”

2)    Again, the use of gout medications for defining gout has limitations, to the extent that some of these medications are used for other purposes, and/or inappropriately. The authors should comment on the possibility that some patients received urate lowering therapy not for gout but for asymptomatic hyperuricemia—a related but different condition.  

We have added reference by Pasina and colleagues describing the proportion of patients with allopurinol Rx that had a diagnosis of gout or other conditions, albeit a small sample (74 patients) from a single pharmacy. Please refer to the paragraph beginning,

“Conversely, not all patients who are prescribed urate-lowering therapies or colchicine have gout….” 

Similarly, some patients receiving colchicine may have been taking it for pseudogout or pericarditis.  The authors should comment in their discussion at least, as to the level of certainty that patients taking gout medications actually have gout. 

We further included the indications suggested by the reviewers. We added comment that we infrequently observed chronic colchicine use in the MESA study. (Same paragraph.)

3)    The possibility that the data presented underestimate the impact of gout on CV disease, specifically because of the possibility that gout treatments may have reduced CV outcomes (together with the fact that untreated gout patients are in the control group), bears additional discussion.  

We have added reference by Krishnan and colleagues on “Furthermore, other authors have reported that untreated gout patients have higher rates of CVD than patients treated for their gout [5].” Along with a short discussion that untreated gout patients in the comparator group may attenuate the gout-CVD association.

In particular, as the authors note multiple large controlled trials in the general CV risk population, and several retrospective studies in the gout population suggest that colchicine may reduce CV risk by roughly 25% or more.

Continuing in that paragraph, we have expanded our discussion that patient treated with colchicine in our treated gout patient group might have their CV risk attenuated based on observations from COLCOT.

The possibility that the authors’ data underestimate gout cardiac risk because the patients are treated should not go unremarked on.  I believe it is possible that a bigger effect would have been seen if the authors could have followed an untreated gout population, including significance in both the male and female populations.

We conclude with the comment that, “Both of these limitations bias the HR estimate towards the null hypothesis.”

4)    The authors’ efforts to correct for other interrelated risk factors are laudable, but it deserves mention that these efforts may have represented in an overcorrection.  For example, while kidney disease is certainly an in independent risk factor for CV disease, multiple studies have suggested the possibility that gout may be a CKD risk factor—so the variables are entangled in complex ways.  It deserves at least brief mention that the correction for some variables may actually remove the effect of mediating pathways through which gout might act to promote cardiovascular disease--again, biasing the study against its actual outcome and perhaps leading to an underrecognition of the impact of gout.

This is a valid point. We agree that removing CKD (and BMI) would strengthen the gout-CVD association (as the crude analysis shows much higher association). We considered adding the discussion about impact of controlling for variables on the causative pathway but ultimately decided against. CKD has clear risk for hyperuricemia and gout (CKD -> gout association). However, increasing data is questioning the gout-CKD causative path with observational studies supporting this association but randomized studies not. Furthermore CKD was not a significant covariate in our analysis. Similar rationale for BMI (which was significantly associated with CKD).

Additional questions:

1)    It would be helpful to specify the frequency/schedule of follow up visits in the study.

These are specified in methods: “Five total examinations, including a baseline, were administered during the time of this analysis: exam 1 (July 2000-July 2002), exam 2 (September 2002-February 2004), exam 3 (March 2004 – September 2005); exam 4 (September 2005 – November 2007), exam 5 (April 2010 – December 2011).”

2)    The authors comment that they saw no differences in outcomes between patients who were compliant and/or on consistent treatment compared with those who were non-compliant and/or on intermittent treatment, and therefore they do not provide outcomes according to these groups.  However, these are potentially important observations; it would be helpful to have at least some data provided.

While we tested but did not find difference between the groups, we don’t feel that MESA is optimal dataset to analyze impact of persistence on CVD, and therefore we did not add more comment.

Other study designs (as example with proportion days covered data) would be better designed to answer questions on persistence.

Reviewer 2

This was a very well-written manuscript describing the risk of CVD in the MESA study participants who were treated for gout. I particularly enjoyed the careful and honest discussion the authors’ had of their limitations and study biases in the discussion. My comments follow:

-The main limitation to the validity of the results is the risk of misclassification. The authors’ appropriately estimate the bias risk on those misclassified as having no gout when they had gout because they were on no gout treatment.

Agreed. We hope that we have added sufficient detail based on comments above.

Another bias that is not discussed are those who got gout medication after being misclassified as having gout.

Please see above response.

Anyone who has practiced clinical rheumatology has seen patients started on allopurinol or colchicine for the wrong reason and there are also a few patients who start colchicine for CPPD disease or other inflammatory arthritis. This scenario would put patients without gout in the gout group and also bias results towards the null.

Agreed, these points were raised by above reviewer, and we hope that comments above are sufficient response.

-In addition, the authors do not discuss the effect of classification of the disease based on the use of medication on the CVD risk under the light that these medications could lower CVD risk (as described in COLCOT) or increase it (as described in the CARES study (White WB, Saag KG, Becker MA, Borer JS, Gorelick PB, Whelton A, et al.Cardiovascular Safety of Febuxostat or Allopurinol in Patients with Gout. N Engl J Med:. 2018;378(13):1200-10.) – although this finding could be controversial in light of the FAST study in Mackenzie IS, Ford I, Nuki G, Hallas J, Hawkey CJ, Webster J, et al. Long-term cardiovascular safety of febuxostat compared with allopurinol in patients with gout (FAST): a multicentre, prospective, randomised, open-label, non-inferiority trial. Lancet. 2020 )

We have included the COLCOT findings in our discussion based on the potential impact of Colchicine use on CVD. As noted above, we included discussion by Krishnan on the risk of untreated patients with gout.

We investigated if Colchicine-only patients differed from ULT patients (with or without colchicine).

“We looked for a differential association with CVD between the colchicine without ULT group and the ULT group (with or without colchicine use) and found none. Both medication groups had 18% CV events.”

The lack of difference between colchicine only and ULT patients may be due to small numbers of colchicine only patient (Type II error).

We have added a line that identification of patients by allopurinol or colchicine use may bias the estimate towards the null based on the above suggested studies,

“As a corollary, identifying patients treated with allopurinol or colchicine could bias the estimate towards the null by selecting patients on medications that may be associated with reduced CV risk based on the Cardiovascular Safety of Febuxostat or Allopurinol in Patients with Gout (CARES) [39] or COLCOT [33] studies respectively.”

-A notable typo is in line 169 where “mail” should be “male”

Noted and fixed.

Reviewer 3

This study examined the risk of CVD amongst patients with gout (defined by medication use) in the MESA study, observing a trend toward an increased risk of CVD in patients with gout which was accentuated amongst female participants. The largest limitation threatening the validity of this study, which I believe the authors recognize and have discussed in their limitations, is the definition of gout status in this study that does not have administrative or clinician diagnoses of gout available, which is where the bulk of my questions arise. Please find my comments below:

  1. As clinical or administrative diagnoses were not available, medication use alone was used to define gout in the study. I have a few clarifying questions in this regard:
  2. Was medication use self-report or confirmed by providers?

MESA medication use is documented by patient questionnaire.

  1. The definition is described as defining gout by use of any medication (including ULT or colchicine). Could a patient be given a diagnosis of gout if prescribed colchicine alone? If so, what percentage of participants were diagnosed in this manner. This raises concern given alternative indications for colchicine.

21% of our sample were colchicine only users (no ULT). (Predominantly intermittent colchicine.) We looked for impact of colchicine only user on CVD. There was no association. Colchicine only users had 18% CV events compared with 18% of ULT users (regardless of colchicine use).

We have added discussion (noted above) about alternative indications for colchicine use.

  1. Unless the cited studies (15-18), it seems like the statement that “this methodology has been utilized by other authors to optimize specificity” is misleading; all of the cited studies included clinical or administrative diagnoses of gout. Have there been any studies that have validated medication use alone in the classification of gout?

Agreed, our original description was inaccurate. We have updated our description.

“Other authors have used medication use as a method to enhance the specificity of ICD claims-based diagnoses [15-19]. With our classification strategy, participants with an (unobserved) diagnosis of gout and not treated with any of the above medications, become part of the larger comparator group.”

  1. It appears that the index date used for analysis was the date of study enrollment in 2000-02, though gout could be defined based on medication use at any of the 4 follow up periods. How was gout status and index date handled in this context? 

We have added greater detail about index data with respect to gout medication use. We did enforce exposure – outcome temporal association excluding patients with CVD preceding gout medication use.

In section 2.2 Identification of participants treated with medications used primarily for gout. We have added the following detail about timing of index date.

“Documentation of the above meds at any of the 5 visits qualified for inclusion. Almost half of participants in our gout medication group were on one of the above medications at visit 1, with similar numbers of first-time documentation of medication exposure across the remaining visits.”

From section 2.3 Clinical End Point Assessments

“To retain a plausible causal pathway, for participants treated with gout medications, we required exposure to these medications to predate CV event.”

From the last paragraph of methods, describing excluded patientss

“…those with documented use of gout medication only after incident CVD event (n = 4)…”

  1. Were data on NSAIDS or glucocorticoids available as colchicine was? These would be important confounders to adjust for, if so.

We did not include NSAID or glucocorticoid data in our analysis. We appreciate the value of the suggestion, but (despite defining our cohort by medication exposure), MESA is not optimized for evaluating pharmaceutical exposure. As we describe, drug exposure is captured at study visits and given the intermittent nature, likely better captures chronic medication use over intermittent use.

  1. If the index date for all patients was the 2000-02 study period, an additional limitation that the authors should comment on would be advances in CVD risk management / prevention over the last two decades. 

Noted and acknowledged in last paragraph of Discussion.

Reviewer 2 Report

This study examined the risk of CVD amongst patients with gout (defined by medication use) in the MESA study, observing a trend toward an increased risk of CVD in patients with gout which was accentuated amongst female participants. The largest limitation threatening the validity of this study, which I believe the authors recognize and have discussed in their limitations, is the definition of gout status in this study that does not have administrative or clinician diagnoses of gout available, which is where the bulk of my questions arise. Please find my comments below:

1.       As clinical or administrative diagnoses were not available, medication use alone was used to define gout in the study. I have a few clarifying questions in this regard:

a.       Was medication use self-report or confirmed by providers?

b.       The definition is described as defining gout by use of any medication (including ULT or colchicine). Could a patient be given a diagnosis of gout if prescribed colchicine alone? If so, what percentage of participants were diagnosed in this manner. This raises concern given alternative indications for colchicine.

c.       Unless the cited studies (15-18), it seems like the statement that “this methodology has been utilized by other authors to optimize specificity” is misleading; all of the cited studies included clinical or administrative diagnoses of gout. Have there been any studies that have validated medication use alone in the classification of gout?

2.       It appears that the index date used for analysis was the date of study enrollment in 2000-02, though gout could be defined based on medication use at any of the 4 follow up periods. How was gout status and index date handled in this context? 

3.       Were data on NSAIDS or glucocorticoids available as colchicine was? These would be important confounders to adjust for, if so.

4.       If the index date for all patients was the 2000-02 study period, an additional limitation that the authors should comment on would be advances in CVD risk management / prevention over the last two decades. 

Author Response

(The authors gave the same response as above.)

Reviewer 3 Report

This was a very well-written manuscript describing the risk of CVD in the MESA study participants who were treated for gout. I particularly enjoyed the careful and honest discussion the authors’ had of their limitations and study biases in the discussion. My comments follow:

-The main limitation to the validity of the results is the risk of misclassification. The authors’ appropriately estimate the bias risk on those misclassified as having no gout when they had gout because they were on no gout treatment. Another bias that is not discussed are those who got gout medication after being misclassified as having gout. Anyone who has practiced clinical rheumatology has seen patients started on allopurinol or colchicine for the wrong reason and there are also a few patients who start colchicine for CPPD disease or other inflammatory arthritis. This scenario would put patients without gout in the gout group and also bias results towards the null.

-In addition, the authors do not discuss the effect of classification of the disease based on the use of medication on the CVD risk under the light that these medications could lower CVD risk (as described in COLCOT) or increase it (as described in the CARES study (White WB, Saag KG, Becker MA, Borer JS, Gorelick PB, Whelton A, et al.Cardiovascular Safety of Febuxostat or Allopurinol in Patients with Gout. N Engl J Med:. 2018;378(13):1200-10.) – although this finding could be controversial in light of the FAST study in Mackenzie IS, Ford I, Nuki G, Hallas J, Hawkey CJ, Webster J, et al. Long-term cardiovascular safety of febuxostat compared with allopurinol in patients with gout (FAST): a multicentre, prospective, randomised, open-label, non-inferiority trial. Lancet. 2020 )

-A notable typo is in line 169 where “mail” should be “male”

Author Response

(The authors gave the same response as above.)

Round 2

Reviewer 1 Report

The authors have adequately responded to my suggestions and concerns.  As noted by me--and the authors--there are certain limitations to their approach but the need to address their question is pressing and their approach is far from unreasonable.  They have adjusted their discussion in ways that I think are informative and actually strengthen their case.

Author Response

We thank the reviewer for their comments and appreciate their insights.

Reviewer 2 Report

Thank you to the authors for their responsiveness to my previous comments. My remaining concern is in regards to the index date definition of the study.

If the authors have defined the index date as the date of the first visit/enrollment into MESA for all participants, then gout status should be defined only at that time point. If wanting to leverage gout diagnoses over follow up, then a matched study defining the index date as the date of gout diagnosis (with matching date for non-gout controls) would be required. It is inappropriate to use a Cox model as described and define exposure/confounding variables using data following the index date. 

Author Response

To address the reviewer’s concern, we have added the following paragraph beginning in line 303 of the manuscript:

“For our multivariate Cox proportional analysis, as participants first exposure to gout medications may have occurred anytime during the study period, their pre-exposure data was left-censored. We did not censor data for the comparator group and therefore their observation period would be longer. If the CV risk changed differentially with time on study this could bias the estimated hazard ratio between groups. This could have biased the result towards the null if CV risk was increasing for the longer the duration in the study and away from the null if the risk was decreasing over time. Additionally, there is potential for immortal time bias due to the fact that study participants in the gout group needed to remain on study long enough to obtain a gout diagnosis.”